# ChainRepair: Enabling Efficient Program Repair with Small Open-Source LLMs

**ACM Reference Format:**
Anonymous Author(s). 2026. ChainRepair: Enabling Efficient Program Repair with Small Open-Source LLMs. In *Proceedings of AIware 2026: 3rd ACM International Conference on AI-powered Software (AIware 2026)*. ACM, New York, NY, USA, 9 pages. https://doi.org/10.1145/nnnnnnn.nnnnnnn

## 1 Abstract

Automated Program Repair (APR) has recently advanced through the adoption of Large Language Models (LLMs). However, state-of-the-art performance typically relies on large-scale proprietary models (e.g., GPT-3.5/4 with 175B+ parameters), limiting accessibility, reproducibility, and cost-efficiency. We present ChainRepair, an autonomous multi-agent APR framework built upon a 7B-parameter open-source model. ChainRepair coordinates five specialized agents through structured chain prompting to enable decomposed reasoning without task-specific fine-tuning. The framework systematically incorporates dynamic execution feedback, grounding repair decisions in empirical evidence rather than static pattern matching. We evaluate ChainRepair on the QuixBugs benchmark, where it repairs 82.5% of defects (33/40) while generating only three candidate patches per bug. Compared to proprietary model baselines, our approach achieves a 25× reduction in model size and a 40× improvement in sample efficiency. These results demonstrate that architectural decomposition and evidence-driven reasoning can substantially mitigate the limitations of smaller open-source models. Our findings highlight a practical and reproducible pathway toward accessible, high-performance LLM-based APR.

## 2 Introduction

The escalating complexity of modern software systems renders manual bug fixing demanding and costly. Automated Program Repair (APR) has emerged as a critical area, evolving from template-based methods [1, 2] to sophisticated learning-based techniques leveraging Large Language Models (LLMs).

While LLMs achieve state-of-the-art code repair [3], their application in leading APR approaches is often monolithic, relying on single, black-box models [4] for the entire process. This strategy struggles with bug heterogeneity. Recent research has thus shifted towards multi-agent systems [5, 6], decomposing the repair pipeline into specialized sub-tasks to enhance transparency and robustness [7].

Despite these architectural advantages, a fundamental barrier remains: the reliance on massive, proprietary LLMs such as GPT-4 [8] and Claude [9]. While powerful, their prohibitive computational costs and closed-source nature hinder reproducibility and raise privacy concerns, rendering them impractical for widespread academic use or integration into CI/CD pipelines. Conversely, accessible open-source models like Mistral (7B) [10] and StarCoder [11] offer a compelling alternative but have historically lagged in raw performance. This performance-accessibility trade-off motivates our central research question:

Can a multi-agent framework built upon smaller, accessible, open-source LLMs achieve repair performance comparable to that of large proprietary models?

We address this question with ChainRepair: an autonomous multi-agent APR framework built on a 7B-parameter open-source model. ChainRepair orchestrates five specialized agents: a Metadata Extractor, Test Case Generator, Test Runner, Analyzer, and Patch Generator. Our work makes the following four core contributions: i) We introduce ChainRepair, a five-agent framework that compensates for smaller model capacity through specialization and structured collaboration, requiring only **3** patch candidates per bug versus 100+ for proprietary model baselines. ii) We demonstrate that 7B-parameter open-source models can achieve competitive APR performance (82.5% success rate on QuixBugs) compared to approaches using 175B+ parameter models, a 25-fold reduction in model size without sacrificing effectiveness. iii) We propose a chain prompting technique specifically engineered to enable sophisticated, multi-step reasoning in smaller models without the need for costly parameter fine-tuning. iv) We design an automated pipeline to integrate rich, dynamic context from code execution and failure symptoms, grounding the model's repair attempts within the limited context windows of smaller models.

Our evaluation on the QuixBugs benchmark confirms the efficacy of ChainRepair. Using LLM-based fault localization and generating only three patch candidates per bug, our framework achieves state-of-the-art performance in a single repair round. These findings validate our approach and offer novel insights into making high-performance, LLM-driven APR both accessible and practical.

This paper is structured as follows. Section 3 introduces a motivating example used throughout the paper to illustrate our approach. Section 4 details the architecture and components of our framework. Section 5 describes our experimental setup and presents and analyzes the results of our evaluation. Section 6 reviews related work, and Section 7 concludes.

## 3 Motivating Example

We illustrate our approach using a bug from the KNAPSACK program in the QuixBugs benchmark [12] (Figure 1). The function implements a dynamic programming solution to compute the maximum value of `items` within a given `capacity`. However, it contains a subtle off-by-one error at Line 11.

```
1   def knapsack(capacity, items):
2       from collections import defaultdict
3       memo = defaultdict(int)
4
5       for i in range(1, len(items) + 1):
6           weight, value = items[i - 1]
7
8           for j in range(1, capacity + 1):
9               memo[i, j] = memo[i - 1, j]
10
11              if weight < j:              # BUG
12                  memo[i, j] = max(
13                      memo[i, j],
14                      value + memo[i - 1, j - weight]
15                  )
16
17      return memo[len(items), capacity]
```

**Figure 1: A buggy version of 'Knapsack.py' program [12]**

The condition $weight < j$ should be $weight \leq j$. This single-character mistake prevents the algorithm from considering items whose weight exactly matches the remaining capacity, leading to suboptimal results. Correcting such a logical error requires understanding the algorithm's intent rather than merely adjusting syntax.

## 3.1 Single-Agent Failure

When provided only with the defective function, a capable open-source model (deepseek-coder-6.7b) incorrectly confirms the program's correctness. It justifies its assessment using standard dynamic programming principles, demonstrating a key failure mode: producing technically sound reasoning for a faulty implementation. This exposes two limitations of the single-agent approach: i) Without code execution, the model lacks empirical feedback and relies solely on static pattern matching. ii) Familiar algorithmic structures bias the model toward assuming correctness, causing it to overlook subtle edge-case errors.

## 3.2 Limitations of Pattern-Matching APR

Current LLM-based APR systems often operate as monolithic, pattern-matching engines. These approaches typically treat repair as a single-step translation task, which leads to several critical failures. We argue that, moving away from monolithic generation toward a decomposed, evidence-driven architecture. This modular approach transforms the repair process into a verifiable chain of distinct analytical steps. A detailed walkthrough of our solution is presented in the following section.

## 4 Proposed Framework : ChainRepair

This section details the architecture of ChainRepair, our multi-agent framework for automated program repair, depicted in Figure 2. The core principle of ChainRepair is to decompose the complex APR task into a pipeline of specialized agents, each performing a distinct, well-defined role. This specialization enhances the quality of each phase, from initial code understanding to final patch validation.

Our framework orchestrates five specialist agents in a multi-phase workflow. The process begins with context gathering (Agents 1-3), where agents parse the code, generate strategic tests, and execute them to produce empirical evidence of failure. This is followed by diagnosis and repair (Agents 4-5), where agents use this evidence to localize the fault and generate candidate patches.

## 4.1 Pre-Repair Analysis and Test Synthesis

The initial phase of our framework is designed to ground the abstract repair task in concrete and empirical evidence. This is accomplished by a three-agent workflow that employs progressively sophisticated prompt strategies. The process begins with structured metadata extraction, advances to creative test case generation, and culminates in the analysis of execution failures. Each agent produces structured output, ensuring a reliable and seamless flow of information to the next stage of the pipeline.

*4.1.1 Metadata Extractor Agent.* The repair pipeline begins with the Metadata Extractor, which builds a semantic understanding of the target code. As detailed in Algorithm 1, it employs a two-pronged approach. First, it parses the source code into an Abstract Syntax Tree (AST) to reliably extract structural information like the function's name and parameters. Second, it leverages this information in a zero-shot system prompt (Figure 3), assigning the LLM the role of a code analysis expert tasked with generating a concise, natural-language explanation of the function's purpose.

---

**Algorithm 1** Metadata Extraction Agent

---

**Require:** $B$: Buggy code
1: Parse $B$ into AST tree $T$
2: function_def ← null
3: **for all** $node$ in AST walk of $T$ **do**
4:     **if** $node$ is FunctionDef **then**
5:         function_def ← $node$
6:         **break**
7:     **end if**
8: **end for**
9: **if** function_def exists **then**
10:     Extract $P = \{p_1, p_2, \ldots, p_k\}$
11:     Extract fname
12:     $\pi \leftarrow$ construct_prompt(fname, $P$, $B$)
13:     desc ← LLM($\pi$) with token limit
14:     $M \leftarrow \{$desc, fname, $P\}$
15: **else**
16:     **Error:** no function definition found
17:     $M \leftarrow \emptyset$
18: **end if**
19: **return** $M$

---

The result is a structured metadata object containing the function's name, its parameter list, and a description of its intended behavior. This object serves as the foundational context that guides all subsequent agents in the repair process.

*4.1.2 Test Case Generator Agent.* This Agent is responsible for crafting a small, high-impact test suite, prioritizing strategic precision over volume. To achieve this, it employs a few-shot prompt, which provides concrete examples to guide the model's output through demonstration rather than just abstract description. As shown in Figure 4, this prompt template combines several components, such as role, task description, concrete examples and structured guidance are provided for three test types: normal operations, boundary conditions, and error scenarios.

Grounded in established testing principles and supported by preliminary experiments, this non-exhaustive categorization balances diagnostic coverage and computational efficiency. This targeted suite is crucial for two reasons: i) It creates a reproducible failure that clearly reveals the bug's manifestation. ii) It provides a comprehensive validation set that helps prevent the generation of overfitted patches later in the pipeline. Here is a walkthrough of

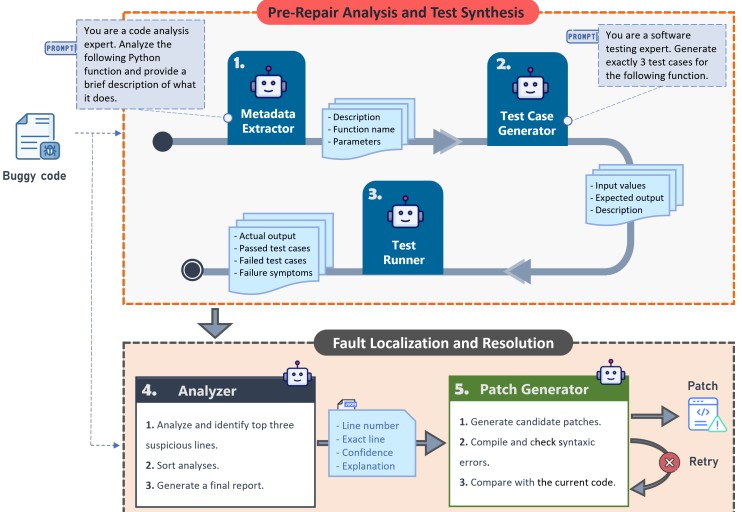

Figure 2: Architecture of the Proposed Framework: ChainRepair

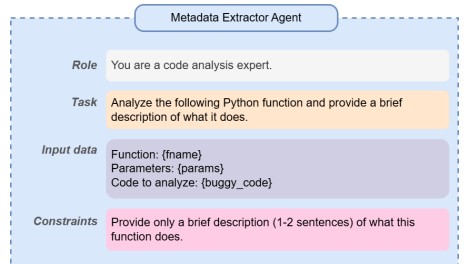

Figure 3: The Metadata Extractor prompt template.

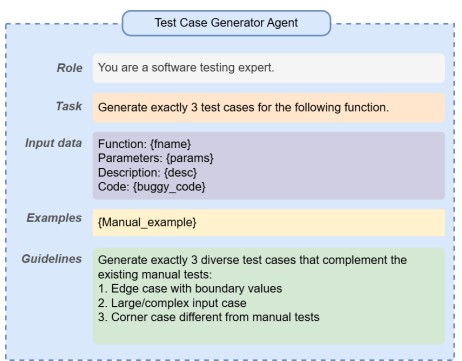

Figure 4: The Test Case Generator prompt template

how the Test Case Generator Agent applies its strategy specifically to our KNAPSACK example: After the first agent determines the function is $knapsack(capacity, items)$, the Test Case Generator's job is to create a few smart tests to see if it works correctly. The generated test cases (in Figure 5) specifically target boundary and edge conditions, such as multiple items with large capacities, minimal capacity, and single-item scenarios.

The objective is not to maximize the number of test cases, but to generate the most strategic ones. These generated cases are used exclusively for fault localization (identifying suspicious lines) and not for final patch validation.

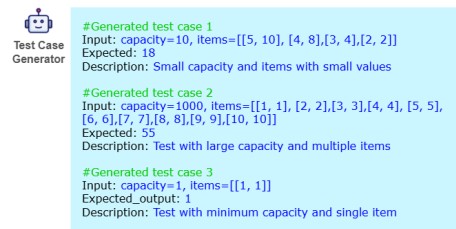

Figure 5: Generated test cases.

*4.1.3 Test Runner Agent.* As LLMs cannot execute code, the Test Runner Agent serves as the crucial bridge between the LLM's abstract analysis and the concrete reality of code execution. It systematically executes the generated test suite, meticulously comparing the actual output against the expected result for each case. Any discrepancy results in a detailed failure report, categorizing the error by type (e.g., TIMEOUT, OUTPUT MISMATCH).

This process transforms a simple pass/fail signal into rich, empirical evidence of the bug's manifestation. This evidence, which provides the verifiable foundation for all subsequent analysis, is encapsulated in a JSON report containing: i) Metadata (function name, parameters, description). ii) Expected and actual behavior. iii) Test results with pass/fail status. iv) Detailed failure symptoms.

## 4.2 Fault Localization and Resolution

In this stage, the system transitions from failure observation to actionable diagnosis and repair. The goal is to accurately identify the root cause of the detected faults and synthesize effective corrective patches. This phase is carried out collaboratively by two agents: Analyzer Agent and Patch Generator Agent.

*4.2.1 Analyzer Agent.* The Analyzer is the diagnostic core of the pipeline, performing LLM-based fault localization. It operates by correlating all available context: the original code, the metadata, and, critically, the detailed failure symptoms provided by the Test Runner. By grounding its analysis in this empirical evidence, the

Analyzer goes beyond traditional static pattern-matching, enabling a more robust and context-aware diagnosis. Rather than pinpointing a single faulty line, the agent produces a ranked list of the top three most likely buggy lines, each accompanied by a calculated confidence score. This strategy offers multiple repair targets and provides native support for multi-location bugs.

As illustrated in Figure 6, the agent uses a multi-context prompt to produce a detailed analysis for each suspicious line, structured as follows: i) LINE_NUMBER: The suspected line number. ii) EX-ACT_LINE: The content of the corresponding line. iii) CONFI-DENCE: The LLM's confidence score (from 0.0 to 1.0). iv) EXPLA-NATION: A detailed rationale explaining why the line is considered suspicious. This approach isolates faults by converting raw diagnostic data into actionable targets for the subsequent repair agents.

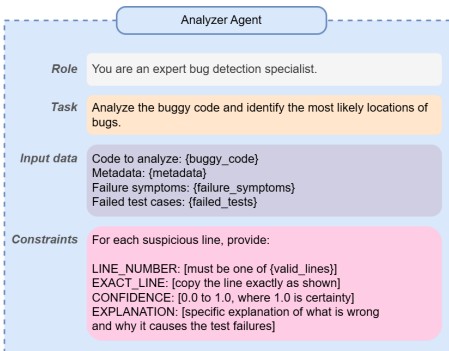

**Figure 6: The Analyzer prompt template**

*4.2.2 Patch Generator Agent.* The Patch Generator is responsible for producing a candidate patch for each suspicious buggy line, which is subsequently tested and validated through prompt engineering, LLM interaction, and multi-layer validation. As illustrated in Algorithm 2, it takes as input the bug analyses provided by the Analyzer, the metadata retrieved by the Metadata Extractor, and the execution results from the Test Runner.

---

**Algorithm 2** Patch Generator Agent

---

**Require:** $B$: Buggy code, Bug analyses, $M$: metadata, $Exec$: execution results
1: $patches \leftarrow \emptyset$
2: $ftests \leftarrow$ extract_failed_tests($Exec$)
3: **for** each $analysis$ in bug_analyses **do**
4:     Extract $lnum$
5:     Extract $bline$
6:     Extract $explan$
7:     $\pi \leftarrow$ construct_prompt($M, B, lnum, bline, explan, ftests$)
8:     corrected_code $\leftarrow$ LLM($\pi$)
9:     **if** is_valid_python(corrected_code) **then**
10:        **if** codes_are_different($B$, corrected_code) **then**
11:           $patch \leftarrow$ create_patch(corrected_code)
12:           $patches$.append($patch$)
13:        **end if**
14:     **else**
15:        $\lambda \leftarrow$ construct_prompt($B, lnum$)
16:        corrected_code $\leftarrow$ retry($\lambda$)
17:     **end if**
18: **end for**
19: **return** patches

---

The Generator follows a three-step process: `initial patch generation`, `syntax validation`, and `similarity check`. The most intricate phase is the patch generation itself, which relies on prompt engineering to synthesize candidate fixes for each identified bug. To accomplish this, the agent leverages two distinct prompts.

The first, an augmented prompt, integrates all relevant bug-related information, such as detailed analyses and failed test cases, alongside the buggy code to guide the LLM during the initial generation phase. If this first attempt fails, a secondary mandatory prompt is triggered as part of a one-time retry mechanism. This refined prompt provides the LLM with explicit and directive instructions on the nature and scope of the required modifications, ensuring that the regenerated patch significantly diverges from the initial output. Finally, each generated patch undergoes syntactic validation to verify that it introduces no new errors and exhibits sufficient structural and semantic variation from the original code, thereby preventing the creation of redundant or duplicate patches.

## 5 Experimentation and Validation

To evaluate our framework, we conducted several experiments designed to answer the following questions:

**RQ1**: How does ChainRepair perform against state-of-the-art APR methods on the QuixBugs benchmark?

**RQ2**: Does the proposed multi-agent prompting framework outperform transferable zero-shot prompting across various LLM backbones?

To answer **RQ1**, we benchmarked ChainRepair on the QuixBugs benchmark [12] and compared its performance against leading LLM-based APR methods. We evaluated four distinct configurations of our framework, each using one of the following lightweight open-source LLMs as its backbone: Mistral-7B, DeepSeek-Coder-7B, Qwen2.5-Coder-7B, and LLaMA-3.1-8B. For each configuration, every generated patch was validated using automated checks. We measured success using a set of starndard metrics.

To answer **RQ2**, we conducted a zero-shot ablation to determine if repair gains stem from model-specific behavior or transferable framework design. This quantifies the value of multi-agent coordination against a single-pass baseline across various LLM backbones.

### 5.1 Experimental setup

We implemented our framework, ChainRepair, in Python and employed LangGraph to orchestrate the multi-agent workflow. We evaluate four LLM backbones under two conditions: (1) the proposed multi-agent framework and (2) the zero-shot single-agent baseline. All other experimental factors including benchmark programs, compilation checks, and test suites are held constant. We utilized four Hugging Face models via the transformers API in inference-only mode without any fine-tuning. To ensure determinism and sufficient length, we set a 0.1 temperature and a 1024-token limit. Experiments ran using float16 on an NVIDIA P100 GPU, with a protocol of one repair round per buggy program.

### 5.2 Metrics

We assess our framework's performance using these metrics:

- **Resolution Rate:** Our primary efficacy metric. This is the percentage of bugs for which the tool generated at least one plausible patch (passes all provided test cases).
- **Exact Match(Correctness):** The percentage of bugs for which the tool generated a correct patch, defined as being character-for-character identical to the human-written, ground-truth solution.

- **Compilation Rate:** The percentage of all generated patches that are syntactically valid and compile successfully. It is the basic viability of the proposed patches. Formally :

$$\text{Compilation Rate} = \frac{N_{\text{comp}}}{N_{\text{gen}}}. \tag{1}$$

Where,
  - $N_{\text{gen}}$ is the total number of generated patches
  - $N_{\text{comp}}$ patches that compile successfully.
- **Generation time:** The average time, in seconds, taken by the five agents to produce one patch.

## 5.3 Baselines

We compare ChainRepair against twelve recent LLM-based APR approaches, detailed in Table 1. A critical distinction of our work is its focus on open-source models; eleven of the twelve baselines rely on large-scale, closed-source LLMs (e.g., GPT-4, Codex), with only one employing an open-source model (CodeBERT). Of these, FixAgent [13] is the most architecturally similar to ChainRepair, as it also employs a multi-agent framework. Consistent with our fine-tune-free methodology, all selected baselines operate without additional training, enabling a fair comparison.

Finally, because APR performance is highly dependent on the fault localization (FL) assumption, we carefully partition our comparison. ChainRepair is evaluated against each baseline only under its corresponding FL setting (i.e., perfect or imperfect).

**Table 1: Studied Baselines**

| Approach | Default LLM | Patches per bug | Perfect Fault Localization | Published |
|---|---|---|---|---|
| AlphaRepair [14] | CodeBERT | 5000 | ✓ | 2022 |
| Prenner et al. [15] | Codex | 5 | ✗ | 2022 |
| Sobania et al. [16] | ChatGPT | 5 | ✗ | 2023 |
| Xia et al. [17] | Codex | 200 | ✓ | 2023 |
| Omari et al. [18] | GPT-3.5-turbo | 3 | ✗ | 2024 |
| FixAgent [13] | GPT-4 | 5 | ✗ | 2024 |
| CodeFixer [19] | GPT-3.5-turbo | 10 | ✓ | 2024 |
| ChatRepair [3] | GPT-3.5-turbo | 100 | ✓ | 2024 |
| ThinkRepair [20] | GPT-3.5-turbo | 125 | ✓ | 2024 |
| Hu et al. [21] | GPT-o1 | 2 | ✗ | 2025 |
| ContrastRepair [22] | GPT-3.5-turbo | 120 | ✗ | 2025 |
| CodeCorrector [23] | GPT-3.5-turbo | 10 | ✗ | 2025 |

## 5.4 Used Dataset

The QuixBugs benchmark [12] is a prominent dataset used for evaluating APR and fault localization techniques. It consists of 40 classic algorithms, each with a buggy and correct version in both Java and Python, accompanied by a minimal test suite. The dataset is diverse, covering 14 distinct defect classes, such as incorrect operators and missing variable references.

For our evaluation, we focus exclusively on the 40 Python implementations, each of which contains a single-line defect. Following standard practice [16], we preprocess the programs by removing all comments, as they can inadvertently leak the correct solution.

## 5.5 Used Open-Source LLMs

The four open-source LLMs selected for this evaluation were chosen to represent a diversity of training data, context window sizes, and code-tuning strategies. Table 2 summarizes these models.

We selected the Instruct variant of **Mistral-7B** [10] as it is specifically fine-tuned for instruction-following, making it safer and more

**Table 2: Properties of the Used Open-Source LLMs**

| Model | Architecture | Context Window | Training Data | Release |
|---|---|---|---|---|
| Mistral-7B-Instruct-v0.3 | Decoder-only | 32K | Not specified | 2024 |
| DeepSeek-Coder-7B-Instruct-v1.5 | Decoder-only | 4K | 2T tokens | 2024 |
| Qwen2.5-Coder-7B-Instruct | Decoder-only | 32K | 5.5T tokens | 2024 |
| LLaMA-3.1-8B-Instruct | Decoder-only | 128K | 15T tokens | 2024 |

reliable for our agentic workflow. **Deepseek-Coder-7B** [24] utilizes a transformer-based architecture tailored for developer tasks. Trained on 2 trillion tokens, it specializes in accurate, context-aware code completion across multiple languages. We used the Instruct-v1.5 variant, selected for its enhanced instruction-following capabilities that are essential for effective APR. **Qwen2.5-Coder-7B** [25] is tailored for code-related tasks, having been pre-trained on a 5.5 trillion token corpus that includes CodeSearchNet and BigQuery dumps. We selected its Instruct variant to leverage its strong performance on code-specific benchmarks. **LLaMA-3.1-8B** [26] was trained on a massive 15 trillion token dataset. We selected the Instruct variant to test a powerful, general-purpose model. Its strong reasoning skills, even without code-specific pre-training, provide an important contrast to the code-native models in our study.

## 5.6 Results

*5.6.1 Backbone Performance Comparison.* We began by evaluating the four selected 7-8B parameter models as the reasoning backbone for ChainRepair. The results, summarized in Table 3, show a clear performance differential. **Qwen2.5-Coder-7B** emerged as the top performer, achieving the highest resolution rate (82.5%) and the most exact matches (32.5%). Critically, it was also the most efficient, with an average generation time of 142 seconds. **DeepSeek-Coder-7B** also demonstrated strong performance, securing the second-highest resolution rate (77.5%). In contrast, **LLaMA-3.1-8B** and **Mistral-7B** lagged significantly, with resolution rates of 62.5% and 27.5%, respectively. LLaMA-3.1 was also the slowest model, at 21 seconds per repair. All models achieved a 100% compilation rate, indicating that the framework consistently generates syntactically valid code.

**Table 3: Empirical Comparison of Repair Performance**

| Backbone | Resolution Rate (%) | #Exact Match (%) | Compilation Rate (%) | Generation Time (s) |
|---|---|---|---|---|
| Qwen2.5 | **82.50** | **32.50** | 100 | **142** |
| DeepSeek-Coder | 77.50 | 27.50 | 100 | 151 |
| LLaMA-3.1 | 62.50 | 25 | 100 | 154 |
| Mistral-7B | 27.50 | 12.50 | 100 | 173 |

Based on its superior accuracy and efficiency, we selected Qwen2.5-Coder-7B as the default backbone for ChainRepair in all subsequent experiments and comparisons.

*5.6.2 Comparison with State-of-the-Art Baselines.* Table 4 contrasts ChainRepair with twelve baselines on the QuixBugs benchmark. Our framework, using the open-source Qwen2.5 backbone, achieves a highly competitive resolution rate, successfully repairing 33 of 40 bugs (82.5%). We include API pricing to provide economic context for proprietary LLM usage. However, cost is not employed as a formal evaluation metric. The most significant finding, however, is the dramatic efficiency with which this is achieved. ChainRepair requires only 3 patch candidates per bug, an order of magnitude

                                                                                                                            

**Table 4: Comparison with State-of-the-Art Baselines**

| Approach | LLM-Backbone | API Pricing ($) | | #Plausible | #Correct | #Patches per bug |
|---|---|---|---|---|---|---|
| | | **Input** | **Output** | | | |
| FixAgent [13] | GPT-4o | 2.50 | 10 | 40 | 40 | 5 |
| CodeFixer [19] | GPT-3.5-Turbo | 0.50 | 1.50 | 40 | 40 | 10 |
| ChatRepair [3] | GPT-3.5-turbo | 0.50 | 1.50 | 40 | 40 | 100 |
| ContrastRepair [22] | GPT-3.5-turbo | 0.50 | 1.50 | 40 | 40 | 120 |
| ThinkRepair [20] | GPT-3.5-turbo | 0.50 | 1.50 | 40 | 40 | 125 |
| CodeCorrector [23] | GPT-4-turbo | 10 | 30 | 39 | 38 | 10 |
| Xia et al. [17] | Codex | - | - | 37 | 37 | 200 |
| AlphaRepair [14] | CodeBERT | Free | Free | 32 | 27 | 5000 |
| Prenner et al. [15] | Codex | - | - | 23 | 23 | 5 |
| Hu et al. [21] | O1-preview | 15 | 60 | 38 | - | 2 |
| Sobania et al. [16] | ChatGPT | 2* | 2* | 31 | - | 5 |
| Omari et al. [18] | GPT-3.5-turbo | 0.50 | 1.50 | 19 | - | 3 |
| **ChainRepair** | Qwen2.5 | Free | Free | **33** | **33** | **3** |

API pricing is reported per 1M tokens, based on official provider documentation and valid as of February 2026. For each backbone model, pricing is consistent across all evaluated approaches. Entries marked with * denote models without separate input/output token pricing

less than most baselines, which require 100 to 5000 candidates (e.g., AlphaRepair [14], ThinkRepair [20]). This highlights a critical trade-off: while several baselines achieve perfect (40/40) scores, they do so using massive, proprietary models (e.g., GPT-4) and extreme sampling rates.

Furthermore, the 100% success rate on the well-established QuixBugs benchmark may raise concerns about data leakage. ChainRepair's 82.5% success rate with a 7B model and minimal sampling provides strong evidence of generalizable reasoning.

Notably, ChainRepair demonstrates a relatively short generation time (142 s) compared with the time required by most APR methods, highlighting the need to consider the trade-off between samples volume and generation efficiency in APR framework design. Furthermore, ChainRepair performs repairs without rely on explicit defect localization, thereby improving its generalizability (Answer to RQ1).

*5.6.3 Efficiency and Cost Analysis.* Beyond patch count, the economic and resource implications of our approach are a key contribution, as shown in Table 5. ChainRepair's multi-agent architecture leads to significant efficiency gains by drastically reducing the required LLM calls per successful repair.

Our ChainRepair framework achieves an 82.5% success rate while generating only 120 total patches with a reduced model size of 7B parameters. Most notably, because it operates with zero cloud-based computational costs, as it can be deployed entirely on local or on-premise infrastructure. This represents a fundamentally different cost-performance paradigm compared to cloud-dependent baselines. When normalized for the 17.5% performance gap relative

**Table 5: Performance-Efficiency Tradeoff Analysis**

| Approach | Size (B) | Success (%) | #Total Patches (40 Bugs) |
|---|---|---|---|
| ThinkRepair [20] | 175 | 100 | 5,000 |
| ChatRepair [3] | 175 | 100 | 4,000 |
| CodeCorrector [23] | 175 | 95 | 400 |
| FixAgent [13] | 1,700 | 100 | 200 |
| **ChainRepair** | 7 | **82.5** | **120** |

to perfect success, ChainRepair demonstrates superior efficiency across multiple dimensions:

- Achieves 96% of the compression ratio (7B vs. 175B-1,700B parameters) while retaining 82.5% effectiveness.
- Generates 40% fewer patches than the next-most-efficient approach (120 vs. 200 patches).
- Removes recurring cloud inference costs entirely, making it viable for resource-constrained environments and practical CI/CD pipeline integration.

While ChainRepair incurs no API costs when deployed locally, computational resources are still required. Organizations must consider hardware costs, energy consumption, and prompt maintenance. However, the one-time hardware investment and absence of per-query API fees offer advantages for high-volume use cases.

## 5.7 Error Analysis: Where ChainRepair Fails

We analyzed the 7 bugs (17.5%) that ChainRepair failed to repair. As summarized in Table 6, these failures are not random but cluster around specific types of complex algorithmic logic.

**Table 6: Analysis of seven Failed QuixBugs Programs**

| Bug Name | Bug Type | Reason for Failure |
|---|---|---|
| TOPOLOGICAL_ORDERING | Logic Error | Requires understanding of graph theory to distinguish between incoming and outgoing node dependencies. |
| LCS_LENGTH | DP Recurrence | The model recognizes the dynamic programming structure but lacks the rule to correctly identify the required cell transition for LCS. |
| NEXT_PALINDROME | Off-by-One | The model failed to reason about palindrome length preservation; the arithmetic relationship between original and result length is too abstract. |

*Note: The remaining 4 bugs follow similar failure patterns involving complex dynamic programming logic, graph invariants, or off-by-one reasoning errors.*

The failures consistently involve bugs that require a deep understanding of *complex data structure invariants* (e.g., in DP or graph algorithms) or *multi-line, non-local logic modification*. This indicates that while the multi-agent architecture excels at focusing the 7B model on single-line corrections with clear empirical feedback (as in KNAPSACK), it still cannot fully compensate for the capacity limitations needed for multi-step, abstract reasoning required by the hardest algorithmic bugs.

Besides, to investigate whether ChainRepair is truly reasoning or simply "memorizing" benchmark solutions, we analyzed the 33 successful repairs. As shown in Table 7, we found that a majority of the patches (60.6%) were semantically equivalent to the ground truth but syntactically different.

**Table 7: Patch Diversity Analysis for 33 Successful Repairs**

| Category | Count | Percentage (%) |
|---|---|---|
| Exact Syntactic Match (Memorization/Retrieval) | 13 | 39.4 |
| Semantically Equivalent, Different Syntax (Reasoning) | 20 | 60.6 |

This finding is significant: it strongly suggests that the multi-agent framework is enabling genuine synthesis and reasoning, not just template-based retrieval. This evidence provides a critical counter-argument to the general threat of data leakage on established benchmarks, demonstrating that the model can derive a correct solution in a novel way.

**Table 8: Empirical Comparison of Repair Performance in Zero-shot settings**

| Backbone | Resolution Rate (%) | #Exact Match (%) | Compilation Rate (%) | Generation Time (s) |
|---|---|---|---|---|
| Qwen2.5 | 57.5 | 17.5 | 100 | 7 |
| DeepSeek-Coder | 30 | 15 | 100 | 14 |
| LLaMA-3.1 | 20 | 15 | 100 | 9 |
| Mistral-7B | 20 | 5 | 100 | 17 |

## 5.8 Prompt Transferability

The result of the conducted experiment using four open-source LLM backbones withing Zero-shot prompting is depicted in Table 8, which shows a critical limitation in APR prompt transferability: while prompts transfer syntactically, their effectiveness degrades sharply across backbones. For instance, the same prompt achieves a 57.5% resolution rate on Qwen2.5, yet drops to 20% on LLaMA-3.1 and Mistral-7B. This discrepancy proves that naïvely transferable prompts do not guarantee transferable performance. In contrast, the multi-agent framework exhibits significantly lower variance across backbones. While absolute performance scales with model capacity, the framework's relative advantage remains consistent. By externalizing reasoning into explicit roles and interactions, the structure compensates for backbone-specific weaknesses rather than relying on a single prompt to elicit complex logic. This finding shifts the focus of prompt transferability: rather than testing if a single prompt generalizes, our results show that structured strategies outperform monolithic prompts. The multi-agent framework functions as a prompt-level abstraction, a reusable "repair protocol" that yields predictable improvements regardless of the underlying LLM. Although multi-agent execution increases response time, the overhead is bounded and consistent, offering a predictable cost–benefit trade-off. In APR, this latency is typically acceptable, as correctness and robustness are prioritized over generation speed.

Our results demonstrate that performance gains are backbone-independent, persist under identical prompts, and offer systematic advantages over monolithic zero-shot prompting. Therefore, our work's primary contribution is not identifying the "best" LLM, but establishing multi-agent prompting as a transferable, repeatable design principle. By decoupling logic from specific model quirks, this framework remains more durable than prompt pairings optimized for a single generation of LLMs (Answer to RQ2).

## 5.9 Discussion: Architectural Insights for AI Engineering

ChainRepair's success with small LLMs stems from its strict adherence to the *Separation of Concern*s principle. By replacing a monolithic task (Fault Localization + Diagnosis + Patching) with five specialized agents, we achieve three architectural benefits: i) **Reduced Cognitive Load:** each 7B agent is only tasked with one sub-problem (e.g., generating tests or ranking lines). This task-specific focus is simpler than the full APR pipeline, optimizing the model's limited reasoning capacity. ii) **Structured Reasoning Chain:** the output of one agent serves as the highly structured, context-specific input for the next. This chain prompting guides the model's "thought process" more reliably than a single, complex instruction. iii) **Empirical Grounding:** the Test Runner/Analyzer V&V loop grounds the LLM reasoning in code execution, preventing

"hallucinations" and forcing a shift from pattern-matching to empirical diagnosis, as demonstrated in the KNAPSACK walkthrough.

Our findings significantly inform LLM-based tool engineering:

- Moving from 175B+ API-dependent models to local 7B models drastically reduces $CO_2$ emissions per repair and eliminates API costs, promoting sustainable AI research.
- The framework enables on-premises deployment, ensuring sensitive code remains local and resolving a major adoption barrier for enterprise users.
- Using open-source models ensures that our research is fully reproducible, addressing the fundamental scientific challenge posed by proprietary black-box models.

## 5.10 Threats to Validity

*5.10.1 Internal Validity.* Potential data leakage from QuixBugs is mitigated by three factors: (1) the threat is shared by all baselines, preserving our comparative architectural findings (2) over 60% of successful patches differ syntactically from the ground truth, suggesting reasoning over memorization, and(3) failures cluster around complex bugs, indicating capacity limits rather than memorization issues. Additionally, generated tests were not validated as they function only for fault localization. Patch correctness remains verified against ground-truth tests.

*5.10.2 External Validity.* Our evaluation focuses exclusively on QuixBugs (40 Python bugs). This limits the generalizability of our findings to real-world scenarios involving multi-file dependencies, complex architectural issues, or other programming languages. We selected QuixBugs for direct comparison with existing state-of-the-art methods (12 baselines). Extending the architecture to other languages requires specific syntax validators and test runners. We plan to evaluate the approach on larger benchmarks like Defects4J and BugsInPy.

*5.10.3 Construct Validity.* We claim to operate without Perfect Fault Localization (PFL), yet our Analyzer agent provides top-3 ranked suspicious lines. We clarify that this is a form of LLM-based fault localization that is automated and integrated into the repair workflow. This is a more realistic and practically useful setting than assuming PFL, but the quality of this localization is dependent on the LLM's diagnostic reasoning capabilities. Our evaluation focuses exclusively on LLM-based APR, intentionally excluding traditional systems. Since prior work [17, 27] already established substantial performance gaps between these paradigms, we focus on the current state of the art in LLM-driven repair.

*5.10.4 Conclusion Validity.* With only 40 samples, the reported performance differences (e.g., 33/40 vs. 31/40) may not be statistically significant. Due to computational constraints, we performed single runs with low temperature (0.1). We focus our conclusions on *comparative trends* and *orders of magnitude improvements in efficiency* (25× model size reduction), which are robust metrics, and will include statistical tests and confidence intervals in future work.

# 6 Related work

## 6.1 Initial Applications of LLMs in APR

Initial applications of LLMs for APR treated the task as a standard text-infilling [14] or sequence-to-sequence problem, revealing emergent abilities such as zero-shot [27] and few-shot learning [17]. These methods demonstrated promise, proving that LLMs could correct certain types of bugs without prior APR-specific training. However, their performance was often inconsistent, as they lacked the necessary contextual awareness of the broader codebase to generate correct and coherent patches.

## 6.2 Context-Enriched and Conversational Repair

This limitation led to contextual prompting strategies based on the insight that bugs rarely exist in isolation; their fixes depend on project-specific context such as structure, API usage patterns, and semantic relationships with other components. Researchers therefore enrich LLM prompts with relevant contextual information.

ChatRepair [3] demonstrates that incorporating compilation errors, test failures, and prior patch feedback into iterative conversations with ChatGPT adds real value. Its short conversational loops with few-shot examples reduce repair attempts and improve success rates, although experiments assume perfect fault localization (PFL). Building on this foundation, ContrastRepair [22] further refines the quality of feedback by including failing and passing tests, runtime error messages, and stack traces in the prompt. This iterative, feedback-driven process achieves promising results with fewer API calls. However, these conversational APR approaches remain constrained by the LLM's context window. Operating under fixed token limits restricts the amount of code and feedback history in a single prompt, and extended repair dialogues require multiple API calls to commercial models, incurring a financial cost.

## 6.3 Multi-Agent APR Frameworks

While single-shot LLM prompting is effective, it has limitations. The repair process is often iterative and requires different types of reasoning—understanding the root cause, considering multiple fix strategies, and validating the change. The idea of using multiple LLM agents for complex tasks is gaining traction in APR. Two recent works, FixAgent [13] and PATCH [5], propose multi-agent frameworks to address this problem. FixAgent adopts a three-level, cognition-inspired hierarchy in which specialized agents perform summarization, fault localization, and patch optimization. It adjusts reasoning effort to bug complexity—conserving resources for simple cases while enabling deep, cross-file coordination for complex ones. This method outperformed many APR tools with fewer samples, but its seven-agent coordination and reliance on external tools increase complexity and may limit generalizability. PATCH [5], inspired by human software workflows, simulates tester, developer, and reviewer roles in a conversational loop to report, diagnose, generate, and verify patches. It achieved empirical gains across simple and complex edits. Yet, multiple LLM calls and iterative loops making it more expensive and slower. Overall, these frameworks largely depend on closed-source LLMs exceeding 100B parameters;

**Table 9: Positioning ChainRepair Against Multi-Agent APR**

| Approach | Base LLM | Size (B) | #Agents | #Patches/Bug |
|---|---|---|---|---|
| FixAgent [13] | GPT-4 | 1700 | 7 | 5 |
| PATCH [5] | GPT-3.5 | 175 | 3 | 120 |
| ChainRepair | Qwen2.5 | 7 | 5 | 3 |

as shown in Table 1, eleven studies relied on GPT-3.5-turbo. Additionally, data leakage—where models may reproduce memorized fixes—remains a persistent validity threat. Although newer datasets are often used to mitigate this issue, it cannot be fully eliminated.

## 6.4 Positioning Our Work Against LLM-based APR Architectures

Our work is most closely related to multi-agent frameworks like FixAgent [13] and PATCH [5]. However, a critical distinction is the underlying engineering philosophy: while these prior works leverage the scale and inherent reasoning power of large proprietary models (GPT-4 and GPT-3.5), ChainRepair is explicitly designed to maximize the utility of smaller, accessible open-source models (7B parameters) through architectural compensation. The core difference is summarized in Table 9. Unlike other multi-agent systems, our framework's novelty lies in proving that **architectural specialization and structured chain-prompting can bridge the performance gap with a 25× smaller model**, enabling significant gains in cost, latency, and data privacy for the APR community.

Our work situates itself at the intersection of these trends. We adopt a multi-agent workflow that composes Lightweight open-source LLMs into specialized roles and tightly couples generation to failing test cases for both guidance and automatic validation. This design addresses three practical gaps: (1) it improves robustness by aggregating diverse candidate proposals; (2) it remains practical and auditable by using open models that can be deployed on-premises; and (3) it leverages failing tests not only as a binary oracle but as structured context to inform generation and targeted refinement.

# 7 Conclusion

We presented ChainRepair, an autonomous multi-agent framework designed to address the critical accessibility and efficiency gap in LLM-based APR. By designing a modular solution built on a 7B-parameter open-source model, ChainRepair can compensate for limited model capacity. Our key findings on the QuixBugs benchmark are: i) 7B models achieve an 82.5% success rate (**33/40** bugs), approaching the 100% rate of 175B+ models. ii) Our architecture achieved **40×** better sample efficiency (**3** patches per bug) and utilized a **25×** smaller model size than leading proprietary model baselines. iii) The multi-agent chain prompting enables reasoning over pattern matching, evidenced by **60.6%** of successful patches being semantically equivalent but syntactically distinct from ground truth. Moreover, ChainRepair enables cost-effective, on-premises APR deployment, resolving major data privacy and reproducibility concerns. By eliminating reliance on expensive commercial APIs, this work provides a practical path for integrating LLMs into software engineering workflows. Future work will focus on evaluation on temporal holdouts and multi-file bugs to definitively prove generalization.

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
