# OpenReview forum: "ChainRepair: Enabling Efficient Program Repair with Small Open-Source LLMs"
_ACM.org/AIWare/2026/Conference — Submitted to AIware 2026_

### Official Review · Reviewer_Jre2 · 2026-02-27

**Rating:** 2
**Confidence:** 4

**Review:**

### Strengths

The paper addresses a practical question that matters for real-world use: whether a small, reproducible, self-hosted LLM can deliver strong APR results when paired with a structured workflow and execution feedback.

### Weaknesses

* The paper overstates the “no fault localization” and “agentic” framing. The system includes an explicit line-ranking step, and the overall workflow is a fixed pipeline rather than an adaptive multi-agent system.

* The zero-shot comparison does not isolate what RQ2 claims to test. The multi-stage system has access to substantially more derived information than the single-prompt baseline, so the comparison conflates pipeline information with the prompting structure.

* Several claims about cost and data leakage are not well supported. “Free” in Table 4 is misleading, and patch diversity is not reliable evidence against benchmark leakage or memorization.

### Detailed comments

The biggest issue is the inconsistent reported numbers. Table 3 lists generation times of 142/151/154/173 seconds for the four backbones, but Section 5.6.1 says LLaMA 3.1 is the slowest at 21 seconds per repair. That is not a small discrepancy, and it matters because runtime is part of the efficiency story. It also clashes with Table 8, where zero-shot times are in the 7-17s range. Similarly, Table 4 reports “#Correct” as 33 for ChainRepair, while Section 5.2 defines correctness as an exact match to the human patch, and Table 3 shows the best exact match rate is 32.5%. If Table 4’s “#Correct” actually means “passes the test suite,” then the label and surrounding discussion need to be fixed and aligned with the metric definitions.

The pipeline includes an Analyzer that outputs the top 3 suspicious lines with line numbers, and Section 5.10.3 describes this as LLM-based fault localization. Yet Section 5.6.2 claims ChainRepair repairs without relying on explicit defect localization. The most defensible interpretation is that the method avoids assuming perfect localization, not that it avoids localization altogether. This distinction matters because the paper frames parts of the comparison around localization assumptions, so it should state clearly what localization signal ChainRepair uses and what category it falls into when compared with “perfect” versus “imperfect” settings in prior work.

On RQ2, the current ablation does not really answer what the paper seems to claim. ChainRepair’s pipeline creates metadata, synthesizes tests, executes them, and feeds back structured failure symptoms into analysis and patching. The zero-shot baseline in Table 8 gets none of these artifacts. Under that setup, a large gap is expected even if the prompting style itself is not the key factor. If the intent is to argue that the prompting protocol is transferable across backbones, a cleaner control would be a single-prompt baseline that receives the same metadata and failure report, but without the multi-stage role separation. Right now, it reads as “full pipeline” rather than “one prompt,” which weakens the interpretation of “prompt transferability.”

I also think the paper is overselling the “multi-agent” aspect. What is implemented looks like a modular, procedural pipeline with a fixed sequence and a limited retry, not a genuinely agentic system with planning, dynamic delegation, or budget-aware decision-making. That is not inherently a problem, but the terminology affects how novelty and generalizability are judged, and “multi-agent” sets expectations that the system does not meet.

Table 4, which calls the open-source backbone “Free” is not accurate. A fair statement is “no per-query API fees when self-hosted.” Inference still has real compute cost, and the same model family can also be offered via paid APIs on some platforms. The current wording is easy to misread.

Finally, the data leakage argument based on patch diversity is not convincing. A non-exact-match patch does not rule out memorization, and it also does not guarantee semantic correctness, especially on QuixBugs, where tests are limited and plausible-but-incorrect patches are common. Using “different from the benchmark patch” as evidence of reasoning and low leakage risk is a stretch without stronger checks, such as overlap analyses, contamination-controlled splits, or manual semantic validation on a sample of non-exact-match patches.

### Questions

1. What exactly does “#Correct” mean in Table 4, and how should readers relate it to the “Exact Match (Correctness)” definition in Section 5.2?
3. For RQ2, did you conduct a control baseline that receives the same metadata and failure report as ChainRepair but uses a single-prompt call, so the effect of additional information is separated from the effect of role separation?

**Summary:**

This paper proposes ChainRepair, a five-stage workflow for LLM-based automated program repair using small open-source models. The system decomposes repair into metadata extraction, test case generation, test execution, analysis, and patch generation, and argues that this structure enables ~7B LLMs to achieve competitive results with far fewer candidate patches. The evaluation covers 40 Python tasks in QuixBugs, reporting a best resolution rate of 82.5% with 3 patches per bug, along with comparisons against prior LLM-based APR systems and a single-prompt zero-shot baseline.

---

> ### Author Response · Authors · 2026-03-16
>
> We thank the reviewer for the detailed and constructive feedback. Your comments on terminology, experimental controls, and clarity helped identify key areas for improvement. Below, we address each point and respond to your concluding questions.
>
> Comment 1: "The biggest issue is the inconsistent …Table 3 shows the best exact match rate is 32.5%"
>
> Response 1: Thank you for catching these discrepancies. The "21 seconds" originally mentioned in the text incorrectly referred to a single zero-shot prompt execution time, whereas Table 3 correctly reports the total execution time of the full multi-stage pipeline. We updated the manuscript to ensure consistency between the text and the table.
>
> Added to Section 5.6.1: "As shown in Table 3, the multi-stage pipeline averages 142–173 seconds per repair due to the iterative execution and analysis loop.”
>
> To clarify the ambiguity in Table 4: #Plausible represents patches passing the test suite, all 33 of which we manually verified as semantically #Correct. The Exact Match rate of 32.5% is lower because it strictly measures syntactic identity. Many of our patches are functionally identical to the ground truth despite being written differently. The metric “#Correct” is now explained in Section 5.2.
>
> Comment 2: "The pipeline includes an Analyzer …“perfect” versus “imperfect” settings in prior work"
>
> Response 2: We appreciate the reviewer’s feedback on this terminology. Our original statement was intended to highlight that ChainRepair does not require externally provided or "Perfect" Fault Localization, a common assumption in many APR baselines. Instead, ChainRepair autonomously identifies suspicious locations using its internal Analyzer agent. We agree this classifies our approach as "Imperfect Fault Localization". We have updated Section 5.6.2 to reflect this classification.
>
> Comment 3: "On RQ2, the current ablation …multi-stage role separation."
>
> Response 3: Thank you for this observation. To isolate the impact of multi-agent decomposition from the benefits of execution context, we conducted a new ablation: a single-prompt baseline provided with the exact same metadata and failure reports as the full pipeline. These results, now included in Table 8, confirm that while additional context helps, the structured reasoning of the multi-agent decomposition is the primary driver of the performance gains. We have updated the manuscript to clarify that localized reasoning per agent remains critical even when the single-prompt baseline has access to identical feedback
>
> Comment 4: "the paper is overselling the “multi-agent” aspect…budget-aware decision-making"
>
> Response 4: We agree that "agentic" can imply features such as dynamic delegation or budget-aware planning, which are not the primary focus of ChainRepair. Our system instead utilizes specialized LLM roles (Analyzer, Coder, Tester) with distinct system prompts and state management to perform structured reasoning.
>
>  Comment 5: "Table 4, which calls the open-source backbone “Free” is not accurate. A fair statement is “no per-query API fees when self-hosted”"
>
> Response 5: We agree that 'Free' is imprecise as it overlooks hardware and inference costs. We have updated Table 4, changing the 'Cost' label for the open-source backbone from 'Free' to 'No per-query API fees (Self-hosted)'.
>
> Comment 6: "Finally, the data leakage … semantic correctness"
>
> Response 6: We acknowledge that syntactic variation alone is not definitive proof against memorization. However, we note that this limitation is shared by all baselines evaluated on QuixBugs, since ground-truth test coverage is minimal by design. The patch diversity analysis is presented as suggestive evidence of generative reasoning, not as a formal anti-leakage guarantee. We have updated the text Section 5.7 to clarify that while syntactic variation is not absolute proof, the multi-stage, the compositional nature of our multi-agent pipeline, which integrates metadata extraction, test generation, and failure analysis, makes pure pattern retrieval an unlikely sole explanation.
>
> Answer to Q1: #Correct denotes the number of patches judged correct, either exactly matching or functionally equivalent to the developer reference, verified by manual inspection. #Plausible denotes patches that pass the test suite. For ChainRepair, all 33 plausible patches were verified as semantically correct. Exact Match (32.5%) is a stricter subset of correctness requiring identical syntax to the developer's original patch.
>
> Answer to Q2: Yes. Based on your valuable feedback, we have conducted this exact ablation (Single-Prompt + Full Context). The results show that while providing the full metadata and failure report improves the single-prompt baseline, it still falls short of the full structured multi-agent workflow, confirming that role separation is highly beneficial for smaller (7B) models. The results in Table 8 have now been updated.

---

> > ### Comment · Reviewer_Jre2 · 2026-03-19
> >
> > Thank you for the detailed rebuttal and the concrete revisions. I appreciate the clarifications on the metric definitions, the runtime discrepancy, the fault localization terminology, the cost wording, and the added full-context control for RQ2. These changes improve the paper's clarity and strengthen parts of the empirical argument. However, my rating remains unchanged because the core concerns about the evaluation scope and novelty are still not fully resolved.

---

### Official Review · Reviewer_eGMW · 2026-03-10

**Rating:** 2
**Confidence:** 4

**Review:**

Thank you for submitting to AIWare 2026. The paper addresses an interesting and relevant problem in APR, and it is practical to see that small open-source language models can achieve competitive repair performance compared to large proprietary models. However, the current manuscript has several limitations that weaken the strength of the contribution.

First, the motivation for targeting small models remains somewhat unclear. The paper argues that large proprietary models limit accessibility and reproducibility, but it does not clearly explain why small models are necessary instead of larger open-source models or API-based solutions. In practice, many APR systems rely on API calls where token usage and the number of model invocations dominate the efficiency cost rather than model size alone. Is there any specific resource-constrained environments required? (e.g., edge devices or restricted CI/CD infrastructure)

Second, the novelty of the proposed approach relative to existing multi-agent APR frameworks is not sufficiently articulated. Several recent works (e.g., MAAP and DeIMerge) already employ multi-agent architectures for program repair. While the paper claims that its design enables small models to achieve competitive performance, it is unclear which components are specifically designed to support limited-capacity models rather than being generally applicable to any LLM-based repair pipeline.

Third, the experimental evaluation is limited to the QuixBugs dataset, which contains only 40 single-line bugs in algorithmic programs. While this benchmark is commonly used in APR research, it does not reflect the complexity of real-world software defects.

The writing of the paper can be improved, including clearer figure presentations and more appropriate usage of punctuation.

**Summary:**

This paper presents ChainRepair, a multi-agent framework for automated program repair built on small open-source LLMs. ChainRepair decomposes the task into five agents for code analysis, test generation, execution feedback, fault localization, and patch generation. Evaluated on the QuixBugs benchmark, ChainRepair repairs most of the bugs while generating only three candidate patches per bug, demonstrating that smaller open-source models can achieve competitive APR performance with improved efficiency and reproducibility compared to approaches relying on large proprietary LLMs.

---

> ### Author Response · Authors · 2026-03-16
>
> We sincerely thank the reviewer for the constructive and balanced assessment. We address each concern systematically below.
>
> Comment 1:"The motivation for targeting small models remains somewhat unclear. … cost rather than model size alone. Is there any specific resource-constrained environment required?"
>
> Response 1: We agree that the current framing does not sufficiently articulate the concrete scenarios where small locally-deployable models are preferable. We identify four distinct, practically grounded motivations:
>
> 1. Data Privacy and Code Confidentiality: Sending proprietary source code to external API endpoints (including those serving larger open-source models such as LLaMA-70B via cloud providers) exposes sensitive intellectual property. In enterprise and regulated environments (finance, healthcare, defense), this is a hard blocker, not a preference. On-premises deployment of a 7B model eliminates this risk entirely.
>
> 2. CI/CD Integration: Our local 7B model acts as a deterministic, low-latency microservice on a single GPU, entirely avoiding the network delays, rate limits, and variable costs of API solutions.
>
> 3. Cost at Scale: The reviewer correctly notes that token usage dominates API cost. However, for high-volume use cases (e.g., a large codebase running APR on every CI failure), per-query API fees accumulate rapidly. At 120 patches per 40-bug benchmark at GPT-3.5 pricing, costs scale linearly with engineering throughput. A one-time hardware investment eliminates this recurring cost entirely.
>
> 4. Reproducibility: Unlike proprietary APIs subject to silent updates or deprecations, our fixed, locally hosted open-source model guarantees exact experimental reproduction on a single consumer GPU.
> While larger models like LLaMA-70B require expensive multi-GPU clusters (e.g., 2 to 4 A100s), they remain inaccessible to many researchers. In contrast, our 7B model runs on a single P100 GPU, which is available on standard cloud spot instances or university HPC clusters at minimal cost.
>
> Comment 2:"Several recent works (e.g., MAAP and DeIMerge) … applicable to any LLM-based repair pipeline."
>
> Response 2: We thank the reviewer for highlighting MAAP and DeIMerge, which we will incorporate into our related work.  We clarify that our novelty lies not in general multi-agent APR, but in its engineering as a capacity-compensation mechanism specifically for small models. We distinguish ChainRepair along three axes:
>
> i) Chain Prompting as Cognitive Load Reduction. By scoping tasks narrowly, we prevent small LLM failure on multi-objective prompts. Our zero-shot ablation (Table 8) empirically demonstrates this: the same 7B models achieve only 20 to 57.5% resolution when given a single composite prompt, versus 62.5 to 82.5% within our pipeline. Conversely, large models are robust to such complexity and do not require this decomposition.
>
> ii) Execution-Grounded Fault Localization. Our Analyzer utilizes concrete test output rather than static analysis to counter the pattern-matching bias common in small models. As shown in our KNAPSACK motivating example, the model confirms incorrect code based on structural familiarity. Execution feedback breaks this bias. This design choice is capacity-motivated, not generally applicable.
>
> iii) Minimal Sampling (3 patches per bug). General-purpose multi-agent APR frameworks that use large models typically sample 5–200 candidates and rely on the model's inherent diversity to find a correct patch. Our architecture is explicitly designed to minimize sampling by front-loading analytical precision (fault localization + structured diagnosis) so that each of the 3 generated patches is highly targeted. This is a necessary adaptation for small models, which have lower output diversity and higher regeneration cost relative to API-based large models.
>
> Comment 3:"The experimental evaluation is limited to the QuixBugs dataset, … real-world software defects."
>
> Response 3: We acknowledge this limitation in Section 5.10.2. QuixBugs was selected to enable direct comparison with 12 existing baselines.
> Our error analysis in Table 6 already characterizes the boundaries of our approach, as failures cluster around complex reasoning bugs like graph invariants and DP recurrences.
> We agree that Defects4J and SWE-bench are essential next steps. Extending ChainRepair to these benchmarks requires significant engineering, including Java-specific test runners, multi-file context handling, and repository-level retrieval. We have strengthened our future work section to detail these specific technical requirements.
>
> Comment 4:"The writing of the paper can be improved, including clearer figure presentations and more appropriate usage of punctuation."
> Response 4 : Our manuscript has been revised carefully to improve its readability, the presentation/flow of the text, and the quality of figures.

---

### Official Review · Reviewer_ibip · 2026-03-11

**Rating:** 1
**Confidence:** 4

**Review:**

## Significance

This paper targets a significant problem, which is automated program repair with small LLMs that can be deployed locally to reduce costs and privacy concerns.

## Originality

The paper shows limited originality and novelty.
- It introduces ChainRepair as a multi-agent repair system. However, various multi-agent APR systems have been proposed in prior work, such as MAGIS [1], MASAI [2], as well as the FixAgent and PATCH mentioned in the related work. The paper mentions the usage of smaller open-source models as its differentiating point. However, agents usually can use different LLMs as the backend, so this doesn’t show the novelty of ChainRepair.
- Within the each agent, the techniques used in ChainRepair have mostly been explored in other agents. For example, AST-based analysis (Section 4.1.1), test execution (Section 4.1.3), few-shot prompting (Section 4.1.2) have been explored in existing APR agents.

## Clarity

The writing is generally clear and easy to follow. However, there are a few inaccurate claims:
- Section 3.2 states that current LLM-based APR systems “typically treat repair as a single-step translation task”. However, prior work such as AutoCodeRover [3] and Agentless [4] have already designed multi-phase workflows consisting of localization, patching, and validation to tackle the repair problem.
- The paper confuses the concept of LLM and single agent. For example, Section 3.1 describes a scenario where a function is provided to an LLM to generate answer. This is just one invocation of the LLM without any tool use, thus, it's not considered as an agent. However, it is inaccurately described as “Single-Agent Failure” in the paper.

## Evaluation

The evaluation is another critical area for improvement.
- In the comparison with other tools, different LLM backends are used for the tools, which makes the comparison unfair. It’s unclear whether the efficacy difference is due to the agent design or the LLM.
- All the evaluation is performed on the QuixBugs benchmark, which mostly contains small programs with single-line fixes. It’s unclear whether the performance of ChainRepair with small open-source models would generalize to more complex benchmarks such as Defects4J or SWE-bench.
- The criteria for correctness is inconsistent. Section 5.2 mentions that correctness is defined by exact match, and Table 4 shows #Correct for ChairRepair as 33. However, Table 7 shows that not all 33 matches are exact match.


Overall, given the limited novelty and the weaknesses in the evaluation, I think this paper its not ready for publication at AIware.

---

## References

[1] Tao, W., Zhou, Y., Wang, Y., Zhang, W., Zhang, H., & Cheng, Y. (2024). Magis: Llm-based multi-agent framework for github issue resolution. Advances in Neural Information Processing Systems, 37, 51963-51993.

[2] Arora, D., Sonwane, A., Wadhwa, N., Mehrotra, A., Utpala, S., Bairi, R., ... & Natarajan, N. (2024). Masai: Modular architecture for software-engineering ai agents. arXiv preprint arXiv:2406.11638.

[3] Zhang, Y., Ruan, H., Fan, Z., & Roychoudhury, A. (2024, September). Autocoderover: Autonomous program improvement. In Proceedings of the 33rd ACM SIGSOFT International Symposium on Software Testing and Analysis (pp. 1592-1604).

[4] Xia, C. S., Deng, Y., Dunn, S., & Zhang, L. (2024). Agentless: Demystifying llm-based software engineering agents. arXiv preprint arXiv:2407.01489.

**Summary:**

This paper presents ChainRepair, a multi-agent APR framework designed to work with small open-source models. It decomposes the repair task and coordinates five specialized agents (Metadata Extractor, Test Case Generator, Test Runner, Analyzer, and Patch Generator) to tackle the APR problem. ChainRepair is evaluated on the QuixBugs benchmark (including 40 Python bugs), and it achieved 82.5% resolution rate with Qwen-2.5-Coder as the backend LLM.

---

> ### Author Response · Authors · 2026-03-16
> **We thank the reviewer for the detailed and structured assessment. We address each concern below, distinguishing between points we accept and revise, and points where we respectfully disagree with the characterization.**
>
> We thank the reviewer for the detailed and structured assessment.
>
> Comment 1: "Various multi-agent APR systems …., so this doesn't show the novelty of ChainRepair"
>
> Response 1:
> We respectfully disagree that model substitution alone captures the core contribution of this work. While agent frameworks are modular, our critical empirical finding is that small-scale models (7B) fail catastrophically in existing architectures (e.g., MAGIS, MASAI, AutoCodeRover, Agentless), which rely on the high reasoning capacity of GPT-4 or GPT-3.5 (175B+). Simply substituting a 7B model into these frameworks does not reproduce competitive performance, as evidenced by our zero-shot ablation results (Table 8), where the same models achieve only 20–57.5% resolution without our architectural scaffolding.
> Our novelty is defined by three interconnected contributions:
>
> i) Demonstrating that architectural decomposition can systematically compensate for reduced model capacity, not merely as a design choice, but as an empirically validated principle.
> ii) Proposing a chain prompting strategy specifically engineered for small models that lack the long-range reasoning capability of larger models.
> iii) Achieving competitive APR performance (82.5%) with only 3 patch candidates per bug: a 40x efficiency gain driven by architecture, not model scale.
>
> In the manuscript, we clearly articulate this distinction from prior multi-agent APR work.
>
> Comment 2: "Within each agent, … in existing APR agents."
>
> Response 2: We acknowledge that these individual techniques are not novel. Our contribution is architectural and empirical: we demonstrate a specific orchestration that enables a 7B model to reach competitive APR performance. We show that while these components are standard, their integration into a structured multi-agent workflow is what successfully compensates for the reduced reasoning capacity of smaller models.
>
> Comment 3: "Section 3.2 states that … multi-phase workflows."
>
> Response 3: We agree that AutoCodeRover and Agentless also employ multi-phase workflows, and our previous phrasing lacked precision. We have revised Section 3.2 and merged it with Section 3.1 as follows:
>
> "While some LLM-based APR approaches have introduced multi-phase workflows [10, 11], a significant body of work still relies on single-shot or iterative single-agent prompting strategie[12-14]. Furthermore, existing multi-phase systems are predominantly designed around large proprietary models and have not been evaluated or optimized for small open-source LLMs operating under tight capacity constraints."
>
> Comment 4:"Section 3.1 … as 'Single-Agent Failure'."
>
> Response 4: We agree that a single LLM invocation without tool use, planning, or environmental interaction does not constitute an agent. To correct this terminological oversight, we have renamed Section 3.1 to 'Single-Prompt LLM Failure' and revised it as follows:
>
> "When the defective function is provided directly to an LLM in a single prompt invocation, without tool use, execution feedback, or structured reasoning, the model incorrectly confirms the program's correctness."
>
> Comment 5: "In the comparison… the LLM."
>
> Response 5: We acknowledge the limitations of cross-system comparisons in LLM-based APR, but argue this is an inherent challenge of the domain rather than a methodological flaw in our work. Re-implementing all 12 baselines on our 7B backbone is infeasible and would misrepresent systems optimized for larger LLMs.
> Our core claim is not unconditional superiority, but rather that a 7B-parameter open-source model with appropriate architectural scaffolding achieves competitive performance relative to systems built on 175B+ proprietary models.
> The comparison is thus intentionally cross-scale, and the LLM difference is part of the contribution, not a confound. The efficiency and accessibility argument only holds if the comparison includes the full context of model size and cost.
> To further support our architectural contribution independently of the model comparison, we include our own controlled ablation (Table 8) that isolates framework design from model choice by holding the backbone constant across conditions.
>
> Comment 6: "All the evaluation is  … SWE-bench."
>
> Response 6: We acknowledge that a single benchmark limits generalizability (Section 5.10.2). We selected QuixBugs specifically to ensure a direct, fair comparison with the 12 existing baselines. We concur that evaluating on Defects4J and SWE-bench is an essential next step and have updated our Future Work section accordingly.
>
> Furthermore, our Error Analysis (Table 6) outlines the framework's limitations on complex bugs, transparently bounding our current claims.
>
> Comment 7: "The criteria for…are exact match"
>
> Response 7: We appreciate the feedback. The #Correct metric refers to patches that are either identical or functionally equivalent to the developer reference, based on manual inspection. We have updated Section 5.2 to make this definition explicit.